# Subpopulations in Strains of *Staphylococcus aureus* Provide Antibiotic Tolerance

**DOI:** 10.3390/antibiotics12020406

**Published:** 2023-02-17

**Authors:** Matipaishe Mashayamombe, Miguel Carda-Diéguez, Alex Mira, Robert Fitridge, Peter S. Zilm, Stephen P. Kidd

**Affiliations:** 1Department of Vascular Surgery, Royal Adelaide Hospital, Adelaide, SA 5000, Australia; 2Discipline of Surgery, Adelaide Medical School, The University of Adelaide, Adelaide, SA 5000, Australia; 3Basil Hetzel Institute for Translational Research, The Queen Elizabeth Hospital, Adelaide, SA 5000, Australia; 4Department of Health and Genomics, Center for Advanced Research in Public Health, FISABIO Institute, 46020 Valencia, Spain; 5School of Health and Welfare, Jönköping University, 551 11 Jönköping, Sweden; 6Adelaide Dental School, The University of Adelaide, Adelaide, SA 5000, Australia; 7Department of Molecular and Biomedical Sciences, School of Biological Sciences, The University of Adelaide, Adelaide, SA 5005, Australia; 8Research Centre for Infectious Disease, The University of Adelaide, Adelaide, SA 5005, Australia; 9Australian Centre for Antimicrobial Resistance Ecology (ACARE), The University of Adelaide, Adelaide, SA 5005, Australia

**Keywords:** *Staphylococcus aureus*, small colony variant (SCV), persister cells

## Abstract

The ability of *Staphylococcus aureus* to colonise different niches across the human body is linked to an adaptable metabolic capability, as well as its ability to persist within specific tissues despite adverse conditions. In many cases, as *S. aureus* proliferates within an anatomical niche, there is an associated pathology. The immune response, together with medical interventions such as antibiotics, often removes the *S. aureus* cells that are causing this disease. However, a common issue in *S. aureus* infections is a relapse of disease. Within infected tissue, *S. aureus* exists as a population of cells, and it adopts a diversity of cell types. In evolutionary biology, the concept of “bet-hedging” has established that even in positive conditions, there are members that arise within a population that would be present as non-beneficial, but if those conditions change, these traits could allow survival. For *S. aureus*, some of these cells within an infection have a reduced fitness, are not rapidly proliferating or are the cause of an active host response and disease, but these do remain even after the disease seems to have been cleared. This is true for persistence against immune responses but also as a continual presence in spite of antibiotic treatment. We propose that the constant arousal of suboptimal populations at any timepoint is a key strategy for *S. aureus* long-term infection and survival. Thus, understanding the molecular basis for this feature could be instrumental to combat persistent infections.

## 1. Introduction

*Staphylococcus aureus* remarkably exists in environments as diverse as companion and livestock animals as well as fomite sites, but also in different tissues in the human body, as a commensal (nose and skin) and as a pathogen (blood, bone, heart and others). It possesses a metabolism that permits growth within the diverse physical and chemical properties of these environmental niches. Additionally, it has a unique capability to fend off the toxic assaults inherent or exogenous to these conditions, and this includes enzymes for detoxifying reactive oxygen species (ROS) and acquiring nutrients and essential elements such as iron (discussed in detail below). Being able to continue to survive and grow in conditions that dramatically change defines the trait of bacterial fitness. On its own, this trait gives *S. aureus* an advantage to persist in diseased tissue despite immune and medical responses. Furthermore, it has added to its skillset a capacity to adopt an unusual range of different cell types (cellular profiles) that avoid stressors. While this has been known for more than a century, it was only the remarkable research by Richard Proctor in the early 1990s that began to reveal an understanding of the persisting *S. aureus* cell types. Specifically, *S. aureus*, usually grown as gold, large colonies, had been noticed from clinical samples as small colony variants (SCV), and Proctor provided the information that showed its metabolic changes driving this switch [1,2,3].

## 2. *S. aureus* and Diabetic Foot Ulcer—Foot Infection

*S. aureus* is notorious for being difficult to clear in skin and soft tissue infection and wounds, thereby resulting in chronic pathology. People with diabetes have a 25% lifetime risk of developing a diabetic foot ulcer (DFU) (wound), with more than 55% of DFUs becoming infected, and therefore linked to a diabetic foot infection (DFI) [4]. Further to this, DFI often leads to osteomyelitis (OM; infection of the bone), which has limited treatment options and often requires local amputation. A complex interaction between the host, pathogen and antimicrobial factors determines the successful healing of diabetic ulcers; however, the high rate (28–35%) of chronic and relapsing (DFU and DFI) cases shows that this pathophysiology (DFU to DFI) is poorly understood. The first stage of this pathology is the development of an ulcer. The healing of the ulcer depends on the host’s ability to induce an active defence against the pathogenic bacteria present in the micro-environment. Bacterial load may be too high for the dysfunctional immunology of the host to overcome, or the bacterial virulence, such as in the case of *S. aureus*, allows the bacteria to continue to survive despite the host response.

Unequivocally, *S. aureus* is the predominant bacterium linked to DFU, as well as the progression to DFI and further complications (especially OM) [5,6]. There are bacterial cell states that complicate diagnosis and treatment. *S. aureus* can adopt a state that is still viable but not actively growing and difficult to culture, and this includes SCVs, but also for these diseases there are the biofilm-forming cells and L-form cells (discussed later). The identification of SCVs within patient tissue is therefore very important in preventing chronic and relapsing disease. Failure to recover SCVs results in major susceptibility reporting errors, as the more resistant component of the infection will not have been reported. In a previous study of 47 samples from patients with type 2 diabetes with DFU infections, SCVs were isolated in 4 samples. Notably, all four had prior exposure to multiple antibiotics (including trimethoprim-sulfamethoxazole) for periods of one or more months [7]. Recently, Fattah and Taha also described the isolation of *Staphylococcus* sp. SCVs in infected diabetic foot ulcers, and 9 isolates of SCVs were detected out of 66 samples. In this study, the SCVs showed resistance to beta-lactam, levofloxacin, gentamycin, rifampicin, methicillin and tetracycline, and reduced susceptibility to tigecycline [8].

Within DFU, hidden *S. aureus* can tolerate treatment regimens and remain present in this tissue, reverting to an active state that then causes chronic disease. These cells can subsequently transit to deeper tissue and cause complicated DFI and OM. This disease progression becomes more difficult to treat for various reasons; for instance, a number of studies have shown that *S. aureus* is able to have an intracellular lifestyle, surviving inside human bone cells: osteoblasts and osteocytes (shown mainly for the *S. aureus* SCV state [9,10], but is likely to also be true for its L-form cell state). Within these host cells, *S. aureus* can remain, existing largely undetected (there has been shown to be a dysfunctional host recognition of these bacteria [10]), allowing long-term survival of the bacteria in the bone cell environment, thereby acting as a reservoir for future (relapsing) infections.

## 3. Iron and *S. aureus* Cell Types

The physical and chemical components of the local environment clearly determine *S. aureus’* growth capacity. Importantly, it can survive by choosing a metabolic pathway that enables its persistence. In this environment, there can also be stressors introduced, generated by the host cellular and immune responses (such as the generation of oxidative stress by reactive oxygen species (ROS) through an oxidative burst) but also from medical intervention—such as antibiotics. There is a necessary link between the metabolic response by *S. aureus*, its stress response and subsequent persistence.

The metabolic pathways by which *S. aureus* can grow are essential in its stress response. An ability to potentially have slow growth, but not be damaged by antimicrobial chemicals, is an important part of its persistence. Proctor (and others) have established and reviewed the metabolic pathways [2], the pathways for the generation of energy during the subsequent SCV development and that are essential to *S. aureus* persistence [11], including defects in electron transport, a decreased tricarboxylic acid (TCA) cycle, as well as changes in RNA processing, CO_2_-dependancy, thymidine-dependency or changes in arginine metabolism (details of these are out of the scope of this review). These may not be conserved or ubiquitous across all SCV cells isolated from clinical settings, or indeed all “persister” cells, but defects in the electron transport chain have been seen in SCVs isolated from patients. These defects in SCV cells also seem to provide *S. aureus* with increased resistance to the oxidative burst (and oxidative stress) by host cells which are part of the antimicrobial processes [12]. There are downregulated metabolic pathways in host-relevant iron-starved conditions that overlap with the known metabolic mutants that are SCV-generating pathways (Figure 1).

There is an added link between the metabolic pathways *S. aureus* uses within its host and its ability to survive an antibiotic assault. Studies on antibiotic-resistant *S. aureus* strains have examined their metabolic adaptation to the host environment, upregulating the TCA cycle [13]. Central metabolism is associated with iron availability (as a cofactor or central element in electron transfer in the respiratory chain or TCA cycle, as well as other pathways) [14], and within the host, one of the environmental factors absolutely necessary for bacteria to overcome is the iron limitation. Additionally, it has been shown that iron is required for the biosynthesis of particular virulence factors or toxins (phenol-soluble modulins (PSM), leucocidins and toxic shock syndrome toxin (TSST)). *S. aureus* has iron acquisition mechanisms that are regulated by the transcription factor *ferric uptake regulator* (Fur) but also an alternative sigma factor [15,16]. Iron is needed for several pathways, including metabolic ones, in *S. aureus*. There have been studies showing that in the iron-restricted host environment, *S. aureus* has reduced virulence due to low expression of specific virulence factors (including haemolysin, capsule, *Staphylococcal* Protein A (*spa*), biofilm formation and surface proteins such as Emp) [17,18]. Intriguingly, recent work using an antibiotic-resistant (to ciprofloxacin) strain revealed that under iron restriction, this strain actually increased its virulence, upregulating various virulence factors, as well as amino acid synthesis and energy storage [19]. This adaptation to the host conditions (iron restriction) further links iron sensing (Fur-regulation), antibiotic resistance and *S. aureus* metabolism. Furthermore, iron limitation can result in overexpression of efflux pumps and thereby directly increase ciprofloxacin resistance, at least [20,21]. Related to this, but with further complexity, in co-culture with *Pseudomonas aeruginosa* (a bacterial species that often co-infects alongside *S. aureus*), the iron-regulating sigma factors in both bacteria not only mediated iron transport but influenced antibiotic resistance [22].

## 4. Bet-Hedging as a Model for SCV Development

Chronic or relapsing pathologies can be defined by the ability of the infecting bacteria to subvert or avoid the antimicrobial processes of medical intervention or the immune response. Previously, we have mentioned the ability of bacteria to adopt a slow or quasi-dormant state, as a means of “hiding”. It is interesting to understand this as either a direct response, for instance against antibiotic stress, or resulting from stochastic genetic events within a small number of cells in a population, that occur even in the absence of antimicrobial stressors (antibiotic or immunity). This theme of bet-hedging has been studied in some microorganisms and reviewed in detail [23]. The evolution of antibiotic adaptation is also a discrete field, described elsewhere [24]. Indeed, some research has been presented that accurately argues that SCVs develop as part of the normal lifecycle of *S. aureus*, with a dynamic, antibiotic-tolerant subpopulation (the SCVs) being generated during cell replication (alongside the active cells), and these can rapidly revert to the parental type in the absence of a selective pressure [25]. This is especially true in the case of *S. aureus*, that has various “cell types” it can adopt, and these can occur concurrently within a population of cells and through various mechanisms. These cells may arise as the bacteria aim to cope with their own intracellular state, such as an imbalance of oxidative stresses [26].

While these cell types have been studied for years, the instability of the SCV phenotype hampers research on the mechanisms of SCV formation and physiology. Research on SCVs is largely based on genetically modified strains with defects in the electron transport chain through mutations in *hemB* [27,28,29], *menD* [29,30,31,32] and *thyA* [33,34,35] that create genetically stable SCV (sSCV). Other models include auxotrophism to CO_2_ [36,37], fatty acids [38,39], chorismite synthesis [40] and selection in gentamicin [28]. There are also transcriptional pathways that are linked to SCV development (SigB in particular [41], but recently also MgrA [26,42], through their own altered expression or mutation). These transcriptional events have pleiotropic impacts across the cell. In this context, what has also been found is that SCV development (also true for persisting cells in other bacterial species) can be a consequence of large genetic and transcriptional changes, including genetic rearrangements [43,44,45].

Many of these experiments, however, are based on laboratory generated SCVs that focus on the effect of a single gene that confers the phenotype. There is no common gene involved in all SCVs, and indeed, there are SCVs for which the pathway to the cell type is unknown [29]. Alternative experiments have been utilised to select for SCV by accounting for time-dependent infection and environmental stresses that better represent the formation of SCV in response to clinical infection. This includes long-term growth in limited growth conditions in continuous culture [42,46], long-term infection and intracellular persistence within osteocytes [10] and serial passage within mice [47]. Our research utilises multiple methods of long-term growth to identify the mechanisms of evolution that allow *S. aureus* to adapt to stressful environments, permitting the stochastically generated cells occurring naturally during replication, the time and the conditions to remain. Throughout this review, it is significant to note that the population of bacterial cells within any environment includes a diverse combination and permutation of cell types, and when mentioning specific types (SCV or other cells, such as the early studies on L-form cells, for instance), within one population of cells there is no single type of these cell states present. This has been demonstrated in vitro but also from clinical scenarios, such as cystic fibrosis [48]. Clinically relevant are SCV cells that revert to their parental and pathogenic type (there is complexity to the signals and mechanisms that drive SCV to revert, potentially involving co-inhabiting bacteria). There is growing research in this area (this is outside the scope of our review). It is worth noting that some research identified suppressor mutants (that rescued SCV to their parental type), and common to this was *srrB*, which itself drives a large regulatory pathway linked to (aerobic/anaerobic) metabolism as well as bacterial responses to host stress (specifically, nitric oxide) [49,50].

One form of bacterial cells that survive intracellularly, and have been extensively studied in various bacterial species, are L-forms [48]. L-forms are bacteria with massively reduced or no cell wall. While they remove their cell wall through reduced or a lack of peptidoglycan, and therefore are unstable, within an osmo-protective/isotonic intracellular lifestyle (within host cells), they can not only be maintained but slowly replicate. In this state, they are not attacked by antimicrobial systems, and indeed antibiotics can induce L-forms (especially beta-lactams). There are variations in L-forms, with some mutations allowing strains to switch back and forth from L-form to non-L-form. For *S. aureus*, L-forms have been studied for a number of years [51,52,53,54,55]. Being able to generate laboratory L-forms in *S. aureus* was highlighted by research in the 1980s, either by mutations or growth conditions (initially inducing stable L-forms even in the presence of cell lysing agents such as lysostaphin or 6-aminonpenicillanic acid [55]). This was extended to clinical situations.

A diversity of cell types—not simply one L-form—was highlighted by a number of morphological and phenotypical types of L-form being isolated from bovine mastitis [53]. These seemed to be induced by antibiotic treatment but with different colony types, including a combination of stable and reverting cells. Also clinically relevant (although still largely demonstrated through in vitro studies) is the often co-infecting *P. aeruginosa,* which produces extracellular enzyme/s that induce *S. aureus* L-forms [52]. While extensive research has been performed on L-forms in other Gram-positive bacteria (and to a lesser extent in Gram-negatives), there are interesting developments in recent research with *S. aureus*. This includes a range of morphological and molecular analyses across *S. aureus* strains. Atomic force microscopy, surface charge analyses and transcriptomics reveal a direct link to L-form formation across different cellular pathways (energy metabolism, RNA synthesis and a downregulation of numerous virulence factors) [56]. There is substantial overlap in the dysfunction in central metabolism that induces SCV and becomes part of what seems to be known about the formation of L-forms. A significant study (not only with *S. aureus*) showed that with antibiotics, there is a switch to the L-form state, which is linked to an increased flux through glycolysis, upregulation of the respiratory chain and an associated increase in ROS [57]. These studies have not been performed with altered iron levels. The complexity here is that in these conditions, there is no growth of L-forms, and in aerobic conditions this would lead to cell death; however, in hypoxic conditions (such as would be relevant within host tissues or during persistent colonisation of specific niches), there is growth of L-forms. Furthermore, the generation of intracellular ROS or the assault from exogenous ROS is known to be a key to the switch to a dormant state [58] and specifically being able to balance the ROS linked to SCV generation [26].

## 5. Phenotypic Plasticity in *S. aureus* Persistent Populations

The understanding of the variation of cell types *S. aureus* can adopt can be combined. There is clearly overlap in how we define SCV cells and other *S. aureus* cell types that have a reduced fitness: L-forms, persister cells and cells in a biofilm (an overview is shown in Table 1). There may be common overarching features (such as a reduced intracellular ATP and increased intracellular ROS) but the molecular or genetic pathways in their development are largely different. In *S. aureus,* persisters seem very similar to SCVs but are distinctive (there are also vastly different persisters that arrest their own growth via the action of toxin–antitoxins). In part, the specific environmental stress (host-generated chemicals, immune response or antibiotic) that may be inducing or selecting for the cell type can have a role in the type of SCV, biofilm or persister that is generated. There is much research showing that *S. aureus* persisters exist as part of a metabolically diverse group of cells within a biofilm. *S. aureus* biofilms are complicated systems, with diverse types of cells, and their development is controlled by the interplay between numerous regulatory pathways with cell–cell interactions and communications that are different between strains (this large field is well-described in other reviews [59,60]).

## 6. SCV and Long-Duration Antibiotic Responses

There are different models to study the phenotypic response by bacteria to antibiotics. In each case, these models attempt to mimic the stress profile being directed onto the bacteria. Within the host and during antibiotic treatment, bacteria are growing and cyclically being stressed. This can be replicated through chemostat-based continuous culture [46,61,62] or batch culture systems with continuous periodic re-culture with added antibiotics, an assay long established as adaptive laboratory evolution (ALE) and used to study bacterial adaptation to antibiotics over time [24,63,64]. Recent work used ALE to study *S. aureus* adaptation to vancomycin, finding evolved strains that diverged in their combination of mutation: there was a diversity of genotypes and phenotypes that had adapted to vancomycin, presenting a breadth of “cell types” [63]. In addition to genes known for vancomycin resistance, there were genes that altered their expression, resulting in a changed growth profile and transcriptional landscape of the adapted strains.

## 7. Genotypic and Phenotypic Changes under Antibiotic Exposure

It is worth noting that, while the data described above clearly show that adaptation can be through general phenotypic changes (largely, growth characteristics), these changes potentially vary considering the strain specificity (this can be the bacterial genetic background or the origin). The phenotypic variations can provide an ability to respond to various stresses. On this basis, we have used ALE to determine how a diabetes-related foot infection (DFI) clinical isolate (Table 2) of *S. aureus* adapts to long-term ciprofloxacin exposure (Appendix A). This *S. aureus* isolate was from a bone infection (*S. aureus* UA-DI-55) isolated from a DFI patient with osteomyelitis undergoing amputation to treat the infected tissue. The patient was treated with intravenous amoxicillin during their hospital stay. From this patient, a shaving of uninfected bone distal from the site of infection was taken and the intracellular bacteria was extracted, from which UA-DI-55 was taken. UA-DI-55 formed a non-stable SCV which had reverted to form a large, non-pigmented colony.

We used limited nutrients with added ciprofloxacin stress over multiple generations (Appendix A). We found that over time, UA-DI-55 developed not only an increased resistance to ciprofloxacin (as indicated by the MIC) but also tolerance to other classes of antibiotics. The parental UA-DI-55 strain conferred strong ciprofloxacin resistance, with a MIC of 250 μg/mL and MBC of 750 μg/mL (Table 3). Additionally, UA-DI-55 also presented with some level of resistance to penicillin (MIC of 250 mg/mL) and gentamicin (MIC of 62 mg/mL), but in each case, the MBC (minimum bactericidal concentration, stopping at 100× MIC—continual growth at this concentration is representative of antibiotic tolerance) was increased in the day 7 and day 20 isolates (Table 3). Specifically, at 100× MIC, there was no growth for the parental strain, but day 7 and day 20 had significant growth (indeed, there was increased CFU/mL at day 20, Figure 2). 

Approximately 5 × 10^8^ cells were inoculated into the start of the ALE, and after the first 24 h of growth the culture had approximately 1 × 10^10^ CFU/mL. The CFU/mL had significant fluctuation in the first 10 days of ALE but began to converge to a constant, steady-state-like concentration. After 10 days, there was a steady decrease in CFU/mL for the remainder of the culture The population switched from the large, non-pigmented forming cell types of the parental UA-DI-55 to cells producing non-pigmented colonies, with variations in the colony size (Figure 2). SCVs were observed sporadically during ALE with less than 5% of the population forming SCV. 

Adaptation to long-term ciprofloxacin and nutritional stress selects for cells with increased resistance and tolerance. Samples from the two timepoints were chosen for further analysis: day 7, where the CFU/mL began to converge towards steady-state growth, and day 20, at the conclusion of ALE. After 7 days of the ALE, the MIC and MBC of ciprofloxacin increased to concentrations beyond 1000 μg/mL, which could not be evaluated (Table 3). Indeed, the absolute growth (the biomass that the bacteria could achieve after 24 h of growth) in the presence of antibiotics also was measured (a feature of tolerance), and this significantly shifted with the day 7 and day 20 isolates (Figure 3).

Considering this unique nature of the phenotypic antibiotic responses, we then investigated the underlying genetic changes (whole genome sequencing was performed on day 1 and day 20 isolates, Table 4). A SNP was detected in *ptsH*, which encodes for a non-specific component of the phosphoenolpyruvate-dependent sugar phosphotransferase system (PTS). This resulted in a duplication mutation which added an additional asparagine to the protein in UA-DI-55-d20. PtsH is involved in the uptake and phosphorylation of carbohydrates and the conversion of phosphoenolpyruvate to pyruvate as a by-product in a reversible five-step process [65].

Furthermore, a SNP was detected in *fur*, a global regulator which senses iron-limited environments. This SNP resulted in a missense mutation (presumptively at amino acid 7) from the non-polar amino acid proline to a positively charged arginine which is within the DNA-binding domain (amino acids 1–85) in UA-DI-55-d20 (Table 4). Upon binding to Fe^2+^ (with iron available in the intracellular environment), Fur is active (Fur-Fe) and binds to its DNA operator targets (Fur boxes) to activate/repress the required genes. The lack of Fe^2+^ (iron-limited environments) results in unbound Fur (apo-Fur) and no binding to its operator sites. Fur-Fe represses genes involved in iron acquisition, biofilm formation and antioxidative stress proteins [66]. It is worth noting the aforementioned links between iron and metabolism and metabolism and cell states, such as SCV. Indeed, as also mentioned, iron starvation is in some ways linked to antibiotic resistance.

Studies in *Escherichia coli* found a knockout of *fur* and the subsequent increase in intracellular iron facilitated the evolution of ciprofloxacin resistance within an in vitro setting [67]. Exposure to ciprofloxacin induces physiological changes, resulting in a hyperactivated electron transport chain which produces superoxides (O_2_^−^). Reduced function of superoxide dismutases increases intracellular O_2_^−^ levels that result in increased intracellular ferrous iron and hydroxyl radicals that increase the rate of oxidative damage-induced mutagenesis. This increased rate of mutagenesis can promote the rate that antibiotic resistances evolve in response to ciprofloxacin [67]. This is not unique to Staphylococci; *Pseudomonas aeruginosa*, for instance, has been shown to develop hypermutator phenotypes in cystic fibrosis patients, which has important implications for persistent infections [68].

PTS-dependent uptake systems are utilised (but not essential) for growth in glucose by *S. aureus*. The loss of PTS uptake of glucose has no significant effects on growth in rich media in either aerobic or anaerobic conditions. Conversely, in conditions of anaerobic growth, the loss of PTS uptake significantly reduced growth [69]. This suggests that PTS uptake is a major pathway for the acquisition of carbohydrates in nutrient-limited conditions, as observed in clinical niches of infection. Indeed, the loss of glucose uptake in a mouse model of infection was associated with a reduced bacterial burden, as *S. aureus* must uptake host carbohydrates to accommodate the increased glycolytic flux [69]. Our data revealed the evolution of UA-DI-55 in adaptation to limited media with ciprofloxacin was associated with a reduction in growth, mediated through changes in sugar transport and iron homeostasis. Once again, this phenotypic change to the cell, creating a slow growth and less fit cell type, actually permits the cells to tolerate stresses that are damaging or lethal to their parental types. This response is generic, not simply tailored to one stressor.

## 8. Conclusions

Various biological systems have been studied in the context of their subpopulations, which possess different traits, such as growth under usual conditions (this includes viruses and yeast) or indeed antibiotic resistance. While bacteria have been known for over 100 years to adopt alternative lifestyles as a means to survive and persist within a host environment, for *S. aureus,* an understanding of the molecular basis for its switch to persistent, quasi-dormant lifestyles (especially SCV) was only developed in the 1990s by Richard Proctor. There is now a known complexity of the SCV cell state and its link to a number of chronic infections in the human host. Our current understanding is leading to an ability to detect these bacteria that reside in human tissue as they create a reservoir for relapsing infections. Further to this potential in the diagnosis, studies are currently advancing in developing new treatments that are not only effective against actively growing bacteria but also their persistent cell types. There is unlikely to be only one molecular and genetic pathway *S. aureus* uses to create SCV (or persister or L-form) cellular states. It is interesting to consider that persister cells, SCVs and other cell types are likely generated during replication and as part of the normal growth cycle of *S. aureus*. However, in the usual, un-stressed condition (certainly in laboratory-based culturing), these less fit cell types would not have the opportunity to grow, being outcompeted by the active, rapidly growing cells in the population. When conditions are adverse for the active cells (such as during an immune response, oxidative stress or antibiotic treatment), the less fit cell types, while having a limited growth rate, are indeed capable of persisting. The future understanding of the diverse cellular switching that allows *S. aureus* to persist will improve the outcomes for numerous infectious diseases and prevent chronic and relapsing infections.

## Figures and Tables

**Figure 1 antibiotics-12-00406-f001:**
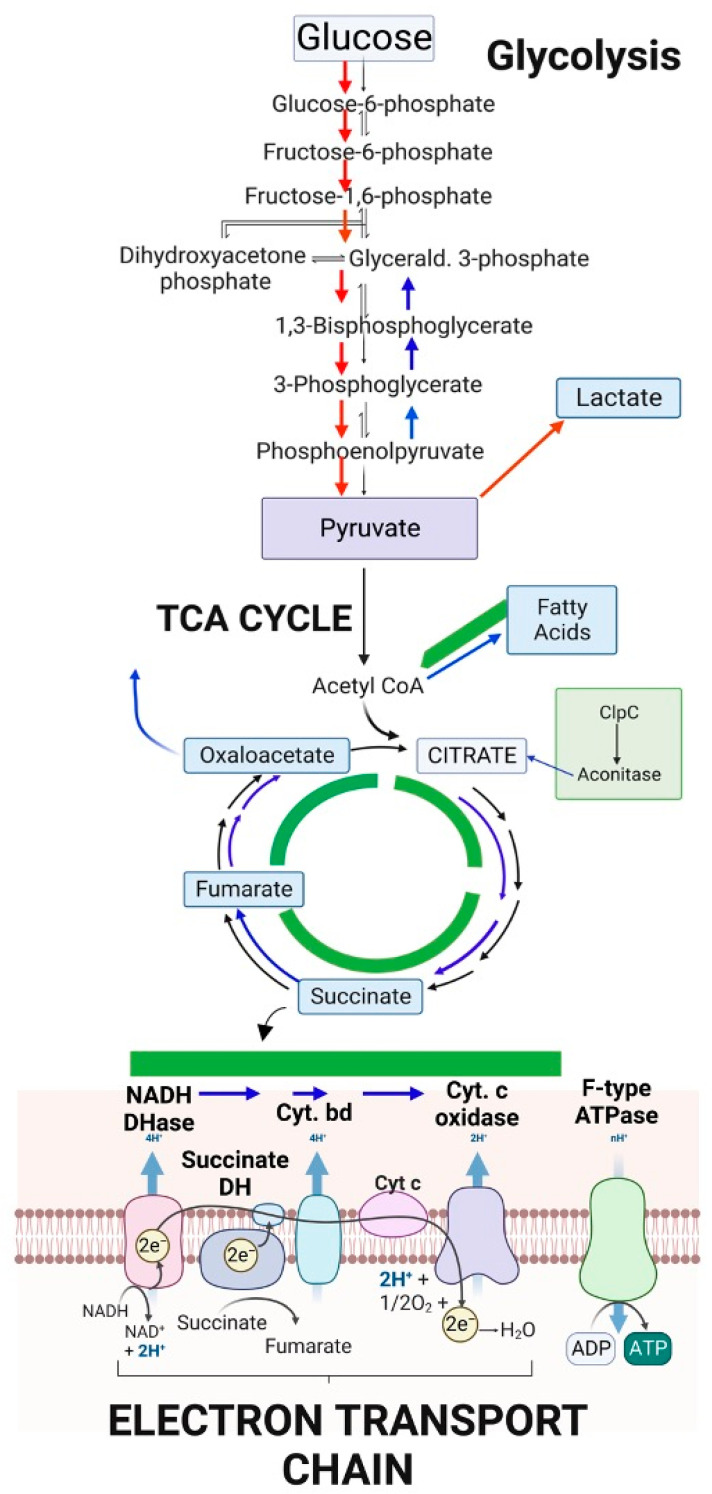
An overview of central metabolism in *S. aureus* and the links to both iron levels and SCV development. Hexose sugars feed into glycolysis, which can be directed into lactate, short-chain fatty acids or to the TCA cycle. The TCA cycle intermediates (NADH and succinate) can then be used in the electron transport chain. Sections of these pathways are upregulated in iron-starved conditions (red arrows) while others are downregulated (blue arrows). There are known metabolic mutants that result in SCV formation, some of which overlap with the iron-regulated pathways (green boxes). This figure was created with Biorender.com.

**Figure 2 antibiotics-12-00406-f002:**
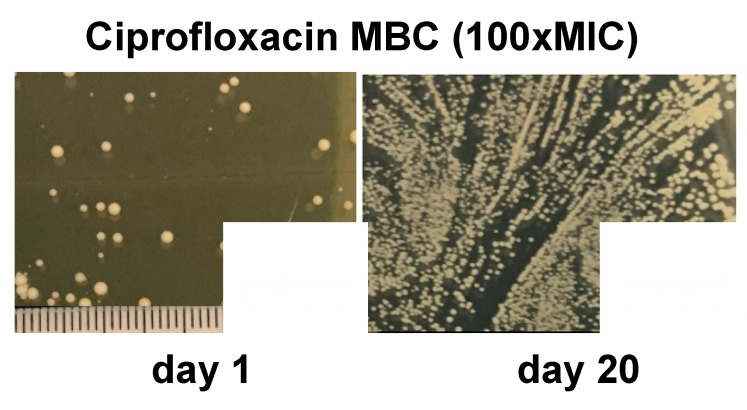
*S. aureus* UA-DI-55 increases its tolerance (measured by increasing MBC) with ciprofloxacin during ALE. The viable colony growth after 24 h on TSA of the strains isolated after 7 and 20 days increases with the 100× MIC concentration of ciprofloxacin.

**Figure 3 antibiotics-12-00406-f003:**
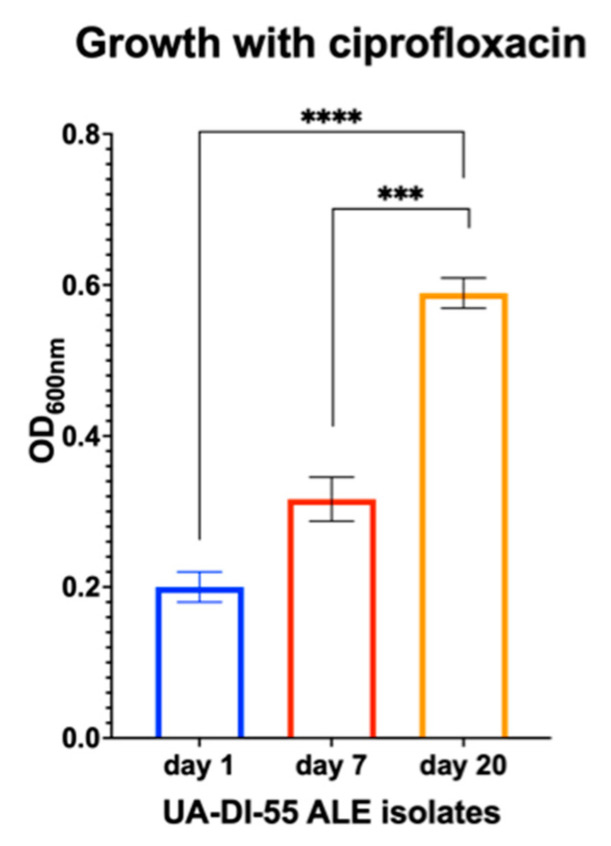
*S. aureus* UA-DI-55 increases its absolute growth with ciprofloxacin during ALE. The growth (shown by OD) after 24 h of the strains isolated after 7 days and 20 days showed significant increases in total biomass with ciprofloxacin. *** *p* < 0.005, **** *p* < 0.0001.

**Table 1 antibiotics-12-00406-t001:** Characteristics of cell types that exist as subpopulations of persistent *Staphylococcus aureus*.

Cell Type	Phenotypic Changes	Genotypic Changes	Metabolic Changes
SCV	Small colony size (<10th normal size) Altered or no pigment Stable SCV Reverting SCV	SNP ^2^ (various genes) Genomic duplications Genomic inversions Gene deletions	Slow growth Low intracellular ATP (defective electron transport chain) Increased intracellular ROS ^1^
L-forms	Reduced or lack of peptidoglycan Slow replication Stable and reverting forms.	To be determined (for *S. aureus*)	Increased respiratory chain Increased intracellular ROS Downregulation of virulence factors Increased glycolysis
Persisters	Antibiotic tolerance	Transcriptional changes SNPs (various genes)	Low intracellular ATP Increased intracellular ROS Damaged TCA
Biofilm cells	Surface structures/proteins Cell–cell interactions Extracellular polymeric substance (EPS)	Complex arrays of transcriptional changes (mainly) Quorum Sensing	The population of cells in a biofilm include persister cells

^1^ ROS—reactive oxygen species. ^2^ SNP—single-nucleotide polymorphism.

**Table 2 antibiotics-12-00406-t002:** Clinical and strain information for the *S. aureus* strain used in ALE.

	UA-DI-55
Colony type	Large
	Non-pigmented
	Stable
Genotype	MRSA
Patient background:	
Age	80 yo
Sex	Female
Condition	Diabetes (10 y)
Infection	Osteomyelitis
Source	Intracellular bone

**Table 3 antibiotics-12-00406-t003:** *S. aureus* strain UA-DI-55 after 20 days of ALE with ciprofloxacin developed a general antibiotic resistance and tolerance. The original strain and the isolates from day 7 and day 20 were assayed for MIC (resistance) and MBC (tolerance) with ciprofloxacin, penicillin and gentamycin.

Strain	Ciprofloxacin		Penicillin		Gentamycin	
	MIC ^1^	MBC	MIC	MBC	MIC	MBC
UA-DI-55	250	750	250	750	62	750
UA-DI-55-d7	>1000	>1000	>1000	>1000	>1000	>1000
UA-DI-55-d20	>1000	>1000	>1000	>1000	>1000	>1000

^1^ Concentrations are in μg/mL.

**Table 4 antibiotics-12-00406-t004:** SNPs detected between the UA-DI-55 day 1 genome and the UA-DI-55 day 20 genome.

Gene	Nucleotide Change	Amino Acid Change	Product
*fur*	20 (G-C)	Arg7Pro	Ferric uptake repressor
hypothetical	475 (A-G)	Lys154Glu	-
*ptsH*	236_238 (dupACG) ^1^	Asp79dup	Phospho-carrier protein
hypothetical	548 (A-T)	Glu183Val	-

^1^ Duplication.

## Data Availability

The datasets generated and/or analysed during the current study are available in the repository BioProject database, with Accession Number: PRJNA821238.

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
