# Peer review of "Subpopulations in Strains of Staphylococcus aureus Provide Antibiotic Tolerance"

_antibiotics, 2023, doi:10.3390/antibiotics12020406_

Round 1

Reviewer 1 Report

Dear authors 

The submitted review contains some significantly important information for the scientific community. I really encourage you to keep up the outstanding work. I have one only comment which is to increase the figures which would elaborate some of the main ideas. 

Author Response

The Reviewer comments are shown with the author responses below these comments and in bold type.

REVIEWER 1

The submitted review contains some significantly important information for the scientific community. I really encourage you to keep up the outstanding work. I have one only comment which is to increase the figures which would elaborate some of the main ideas. 

We thank the Reviewer for their positive and encouraging comments. We have added in a significant, extra figure to elaborate on the interconnections of SCV and metabolism, and iron and metabolism and thereby the overlap of iron and SCV development.

Reviewer 2 Report

Subpopulations in Strains of Staphylococcus aureus Provide Antibiotic Tolerance

This is a review aimed at the understanding of the molecular basis for Staphylococcus aureus's ability to constantly give rise to suboptimal populations as a key strategy long-term infection and survival. These subpopulations, that arise even in positive conditions would be present as non-beneficial, but if those conditions change these traits could allow survival where the original, beneficial population, could not survive. Furthermore, the review suggests that this feature could be instrumental to combat persistent infections of S. aureus.

Major comments

Some general text editing is in order through the manuscript. The text is not badly written just slightly incongruous at times. Just one example, line 348 " the molecular basis for its switch to persisten… was only developed in the 1990s by Richard Proctor."

Figure S2 would be better displayed in the text and not in the supplementary information.

Line 594: "Whole genome sequencing…performed by FISABIO University" – the methods used for WGS construction and annotation should be stated clearly. The service provider should be able to provide you with the necessary data.

As regards to the conclusions, I personally feel that a reference to the similar behavior of viruses (that tend to have sub-optimal populations) would enrich the text further and support your conclusion. I believe that other such situation, such as heteroresistance to antibiotics in bacteria (especially as you do discuss resistance throughout the text) and candida may also apply and strengthen your conclusion.

Minor comment

Table 1: I am not sure that the phenotypes "Stable SCV Reverting SCV" are clear here. This should be rephrased for clarity sake

Author Response

The Reviewer comments are shown with the author responses below these comments and in bold type.

REVIEWER 2

This is a review aimed at the understanding of the molecular basis for Staphylococcus aureus's ability to constantly give rise to suboptimal populations as a key strategy long-term infection and survival. These subpopulations, that arise even in positive conditions would be present as non-beneficial, but if those conditions change these traits could allow survival where the original, beneficial population, could not survive. Furthermore, the review suggests that this feature could be instrumental to combat persistent infections of S. aureus.

Major comments

Some general text editing is in order through the manuscript. The text is not badly written just slightly incongruous at times. Just one example, line 348 " the molecular basis for its switch to persisten… was only developed in the 1990s by Richard Proctor." 

We thank Reviewer for their thorough review and appreciate their accurate suggestions and comments. We have reviewed the text to ensure the reading is clear (this can be seen throughout the “tracked changes” version of the revised manuscript).

Figure S2 would be better displayed in the text and not in the supplementary information.

We were indeed unsure of the most valuable position for this figure and acknowledge the suggestion from this reviewer. As such we have re-worded some of the associated text and now have this figure in the body of the manuscript.

Line 594: "Whole genome sequencing…performed by FISABIO University" – the methods used for WGS construction and annotation should be stated clearly. The service provider should be able to provide you with the necessary data.

We have added the required details in the methods section.

As regards to the conclusions, I personally feel that a reference to the similar behavior of viruses (that tend to have sub-optimal populations) would enrich the text further and support your conclusion. I believe that other such situation, such as heteroresistance to antibiotics in bacteria (especially as you do discuss resistance throughout the text) and candida may also apply and strengthen your conclusion.

This is a very good comment and would, indeed, be the basis for a significant, further discussion on “bet-hedging across biological systems”. We have indicated this in comments we have variously made on the field of evolutionary biology but we are conscious that we do not want to superficially introduce concepts that we do not have the scale and scope in this review article to satisfactorily discuss to the required depth.

In any case, we have added some text, as suggested, to the beginning of the conclusion referring to other biological systems. We felt this was enough to highlight the point made by the reviewer. This is a large field (when starting to consider cell type diversity and sub-populations that exist in isogenic cells as seen across biological systems – viruses, yeast and more, including various bacterial species). A discussion of this to any acceptable details cannot be made in a short way.

Reviewer 3 Report

1.     Additionally, it has a unique capability to fend off the toxic assaults inherent or exogenous to these conditions--- Add  major mechanisms for resistance or survival in tissues.

2.     What are the metabolic changes adapted by S. aureus in different niches as per the title of the manuscript?

3.     Please check the word---showed resistance to------ beta-lactamase, OR is it beta lactum antibiotics.

4.     In the line -SCVs showed resistance to beta-lactamase, levofloxacin, gentamycin, rifampicin, and tet- ….. Methicillin-resistant should also be part of the description.

5.     Explore the line: This disease's 86-year progression becomes more difficult to treat for various reasons.. …

6.     Add the significant description as described by --reviewed the metabolic pathways ----name the major metabolic pathways and key components and their inhibitors.

7.     Is any spefic way is utilized for iron by s aureus for enegy generation. Describe the role of iron requirement for the pathogen.

8.     It will be very good if author add a figure mentioning key target components in iron metabolism or ETC - How the respiratarory chain components will be helpful to overcome the infection or health problem mediated by s aureus.

9.     Please specify the numerous virulence factors)-----

10.  Please mention is Only L type cell utilizing Iron,,, and  associated with--increased flux through glycolysis, up-regulation of the respiratory chain and  increase in ROS.   

Author Response

The Reviewer comments are shown with the author responses below these comments and in bold type.

 REVIEWER 3

  1. Additionally, it has a unique capability to fend off the toxic assaults inherent or exogenous to these conditions---Add  major mechanisms for resistance or survival in tissues.

We thank this Reviewer for these comments. We have added text specifically to this sentence to address this comment. In addition, it should be noted, the core mechanisms are indeed discussed in detail later in the review (ie ROS, iron). 

  1. What are the metabolic changes adapted by S. aureus in different niches as per the title of the manuscript?

We acknowledge this good question from the Reviewer. In the body of the manuscript some words have been added around the reference to Table 1 to ensure it is obvious that we are referring to specific parts of the metabolism of S. aureus. We have added a figure which includes information on the metabolic pathways that are altered in SCV cell states. This is a general switch to a lower growth through changes described for L-forms, persisters and SCVs – this is indeed displayed in Table 1 (the final column title “Metabolic changes”).

  1. Please check the word---showed resistance to------ beta-lactamase, OR is it beta lactum antibiotics.

This change has been made.

  1. In the line -SCVs showed resistance to beta-lactamase, levofloxacin, gentamycin, rifampicin, and tet- …..Methicillin-resistant should also be part of the description.

This change has been made

  1. Explore the line: This disease's 86-year progression becomes more difficult to treat for various reasons.. …

This change has been made

  1. Add the significant description as described by --reviewed the metabolic pathways ----name the major metabolic pathways and key components and their inhibitors.

We thank the Reviewer for this comment. It is indeed important to be specific on the metabolic pathways and we have added the text required. Although the depth of the research in this area is outside the scope and our intention in the wording is that readers can follow this interesting but separate research area in the referenced review articles (as cited).

  1. Is any spefic way is utilized for iron by s aureus for enegy generation. Describe the role of iron requirement for the pathogen.

We thank the Reviewer for highlighting this lack of clarity in this topic within our original manuscript. We have indeed now added some text (seen in the “tracked changes” version of our revised manuscript) but importantly we have also added in a figure that is clear in this area.

  1. It will be very good if author add a figure mentioning key target components in iron metabolism or ETC - How the respiratarory chain components will be helpful to overcome the infection or health problem mediated by s aureus.

We have added a figure that shows the interplay known between the central metabolism for energy production in S. aureus, iron and SCV development.

  1. Please specify the numerous virulence factors)-----

These have now been listed.

  1. Please mention is Only L type cell utilizing Iron,,, and  associated with--increased flux through glycolysis, up-regulation of the respiratory chain and  increase in ROS.   

We thank the Reviewer for this comment and can see our original manuscript lacked clarity in how we were describing this section. For this part of the discussion on L-forms we have added some wording.  These studies have not been performed with the premise of iron changing concentrations.

Round 2

Reviewer 3 Report

All comments have been incorporated by the authors and satisfied.